# Camel (*Camelus* spp.) Urine Bioactivity and Metabolome: A Systematic Review of Knowledge Gaps, Advances, and Directions for Future Research

**DOI:** 10.3390/ijms232315024

**Published:** 2022-11-30

**Authors:** Carlos Iglesias Pastrana, Juan Vicente Delgado Bermejo, Maria Noemi Sgobba, Francisco Javier Navas González, Lorenzo Guerra, Diana C. G. A. Pinto, Ana M. Gil, Iola F. Duarte, Giovanni Lentini, Elena Ciani

**Affiliations:** 1Department of Genetics, Faculty of Veterinary Sciences, University of Córdoba, 14014 Córdoba, Spain; 2Department of Biosciences, Biotechnologies and Biopharmaceutics, Faculty of Veterinary Sciences, University of Bari ‘Aldo Moro’, 70125 Bari, Italy; 3Andalusian Institute of Agricultural and Fisheries Research and Training (IFAPA), Centro Alameda del Obispo, Alameda del Obispo, 14004 Córdoba, Spain; 4LAQV-REQUIMTE, Department of Chemistry, University of Aveiro, 3810-193 Aveiro, Portugal; 5Department of Chemistry, CICECO Aveiro Institute of Materials, University of Aveiro, 3810-193 Aveiro, Portugal; 6Department of Pharmacy-Drug Sciences, University of Bari “Aldo Moro”, 70125 Bari, Italy

**Keywords:** dromedary, active principles, metabolomics, metabolomic profile, standard operating procedures

## Abstract

Up to the present day, studies on the therapeutic properties of camel (*Camelus* spp.) urine and the detailed characterization of its metabolomic profile are scarce and often unrelated. Information on inter individual variability is noticeably limited, and there is a wide divergence across studies regarding the methods for sample storage, pre-processing, and extract derivatization for metabolomic analysis. Additionally, medium osmolarity is not experimentally adjusted prior to bioactivity assays. In this scenario, the methodological standardization and interdisciplinary approach of such processes will strengthen the interpretation, repeatability, and replicability of the empirical results on the compounds with bioactive properties present in camel urine. Furthermore, sample enlargement would also permit the evaluation of camel urine’s intra- and interindividual variability in terms of chemical composition, bioactive effects, and efficacy, while it may also permit researchers to discriminate potential animal-intrinsic and extrinsic conditioning factors. Altogether, the results would help to evaluate the role of camel urine as a natural source for the identification and extraction of specific novel bioactive substances that may deserve isolated chemical and pharmacognostic investigations through preclinical tests to determine their biological activity and the suitability of their safety profile for their potential inclusion in therapeutic formulas for improving human and animal health.

## 1. Introduction

Urine utilization (either consumed or locally applied) has attracted the interest of health academicians and intellectuals since ancient times due to the widespread belief of the many preventive and curative potentialities of this biofluid as a treatment against several ailments [1]. Despite the controversial nature of the topic, the availability of worldwide references from current and ancient civilizations [2,3] across cultures and societies is remarkable. However, a systematic review of its bioactivity, metabolomic profiling, key biochemical parameters for pharmacological research and drug design, and repercussions on public health has not been performed to date as it has for the controversial utilization of other biological substrates by humans [4,5,6,7,8]. Similar to other bioactive substances of animal origin, further standardized research is needed for the optimization of the process of their isolation as well as high-quality clinical trials to evaluate their efficacy and the safety of their use for the treatment of certain diseases [9].

The 5000-year-old document ‘Damar Tantra’ is the oldest known medical treatise to first mention the therapeutical use of human urine (‘Shivambu Kalpa Vidhi’) in the Indian ayurvedic tradition. Among other prescriptions, this miscellanea recommended that one should drink one’s own urine in the morning for general revitalization, prophylaxis, and as an adjunct to conventional treatments [10]. Similarly, historical records on the use of camel urine for medical purposes date back to a text from 1020 titled The Canon of Medicine by Avicenna, a Persian Muslim scientist who authored numerous globally acknowledged scientific papers and medical books [11]. Urine consumption was not exclusively practiced by oriental cultures [12,13]; urine would also appear cited in Greco-Roman classical medical texts (13–20th centuries BC) in which it was referred both as a remedy and diagnostic tool [14,15,16,17].

Multiple in vivo research undertaken since the early 1990s have evidenced the effects that urotherapy (both with autogenous or heterogeneous urinary extracts) has on different pathological processes, such as urinary infections, gonarthritis, desensitization, endocrinological problems, migraine, pruritus, asthma, urticaria, eczema, psoriasis, acute and subacute glomerulonephritis, experimental ulcers, lymphoid depletion in intestinal segments, the induction of thrombocytosis in peripheral blood and megakaryocytosis in the spleen, and the neutralization of the bone marrow colony-stimulating factor [18]. Some descriptive studies also referred to this alternative practice as a remedy for the treatment of many human diseases such as abdominal tumors, gastro-intestinal disorders, and other chronic conditions with the urine of different animal species, including small and large domestic ruminants, llamas, camels, buffaloes, elephants, and equids [12,19,20]. In the contemporary scene, Armstrong [21] would praise the value of urine as a polyvalent curative agent by comparing its bioactivity with the natural occurrence of organic composting. Subsequent research has made revolutionary findings in relation to the potentialities of urotherapy for patients with cancer, which is a leading global health issue [22,23]. However, camel urine’s alkalinity (high levels of potassium and magnesium, albuminous protein, and low concentrations of uric acid, sodium, and creatine), unlike other animal urine, may be the source for the comparatively higher historical prevalence of its use [24]. Concretely, high alkalinity is the property to which other human medicines’ desirable properties such as antimicrobial potential activity has been ascribed [25].

In particular, drinking camel urine alone or mixed with milk is a widely renowned practice within the scope of folk medicine [26]. This habit has frequently been described in Arabian countries, especially among Bedouin tribes with a strong heritage from the Muslim religion [27,28], among which the urine of virgin female camels is the most appreciated [29]. In line with the aforementioned uses, Gole and Hamido [30] contextualize their review on the fact that despite its traditional use as medicine, there is no scientific dosage at which camel urine can be applied as a medicine for different diseases, and the methods for camel urine’s formulation and utilization for the care of patients vary internationally. Furthermore, these authors suggest that there is a need for the biochemical composition of camel urine to be scientifically extracted and formulated as a therapy to prevent its raw consumption and, consequently, its negative effects on human health [30].

Recently, several in vitro and in vivo studies have explored the biological effects of camel urine, namely, its bioactivity towards a range of tumoral and non-tumoral cells [31]. However, information on the bioactivity of specific urine metabolites, their potential interactions, and safety profiles is very scarce. Thus, further investigation is needed to perform a detailed characterization of the chemical composition of this biofluid and to assess how its metabolic profile and properties are affected by different extrinsic and intrinsic factors. As stated by Kumar et al. [32], urine is constituted by different secretions that aid the protection of the urogenital tract against environmental pathogens/threats and organic failures. Once the biological activity of whole urine is tested and confirmed under experimental conditions based on standardized principles established through international conventions, the proceeding steps involve the characterization of its chemical composition and the subsequent isolated prediction and testing of the bioactivity, through in-silico and in vitro/in vivo experiments, respectively, of each of the molecules encountered to classify them into functional groups [33,34]. This would help to rationalize and expand the potential value of camel urine as a natural substrate for the discovery of novel bioactive substances with applications in biomedicine. Furthermore, the isolation and testing of new bioactive compounds may also enhance the opportunities for their application in a range of other industrial fields such as the synthesis of nanomaterials and the generation of hydrogen [35].

In this context, the identification and quantification of the low-molecular-weight (<1 kDa) small molecules present in urine as a biological system, based on metabolomic technologies, may constitute a powerful tool for the detailed characterization of urine metabolomic profiles and in turn individual phenotypes [36,37]. By using complementary analytical methodologies such as gas chromatography–mass spectrometry (GC-MS), liquid chromatography–mass spectrometry (LC–MS), and nuclear magnetic resonance (NMR) spectroscopy, metabolomics enables a large number of metabolites to be detected simultaneously, enhancing the possibility of identifying new biologically active compounds [38]. However, given the expected complexity and variability of biological samples, it is of paramount importance to ensure that standardized protocols are in place for the samples’ handling and processing so as to avoid undesirable sources of variation [39,40].

While previous review articles on camel urine’s therapeutic benefits have mainly summarized the bioactivity test results conducted by separate studies [13,24,27,31,41], this review aims to offer a comprehensive view of the current knowledge on the camel urine metabolome and its biological effects, as well as highlight the main knowledge gaps and research challenges around the topic that need to be addressed for a contrasted and safe use of this animal-derived substrate in drug discovery focused on public health [42]. A detailed outline of the camel urine metabolome is provided herein based on the nature and uses of each molecule reported to be present in this biofluid, thereby enforcing the existing body of scientific literature aimed at describing the therapeutic role of any specific components in urine. Additionally, the implications of different pre-processing and analytical methods in the metabolomics research outcomes are discussed and recommendations on sampling methodologies that could improve ongoing industrial and biomedical applied research are issued. Afterwards, this review contributes to the pioneering creation of a repository for camel urine metabolomics and related metadata, and encourages the integration of documented clinical experiences and experiential observations through transdisciplinary, robust preclinical and clinical research that aids the definition of new drug candidates (reverse pharmacology) [43] that are contained in camel urine. All in all, this functionally multidisciplinary research will promote the sustainable conservation of traditional medicines derived from autochthonous animal populations in line with the 2003 UNESCO Convention on the Safeguarding of the Intangible Cultural Heritage.

## 2. Methods

The present review was carried out in four stages, namely, the definition of the most relevant topics to be included in the review, database generation and pruning, document content evaluation, and the performance of a complementary review on camel urine’s chemical composition. PRISMA 2020 guidelines were followed to conduct the present systematic review [44].

### 2.1. Literature Search Strategy and Exclusion Criteria

#### 2.1.1. Search Repositories

According to the criteria defined for the selection of peer-reviewed literature platforms to be used for the compilation of published research [45,46,47], ScienceDirect (www.sciencedirect.com) was selected as the primary search system used in our study. However, since only one paper addressing the camel urine metabolome was indexed in this platform and considering that Booth et al. [48] proposed to include an average of 8–12 studies to effectively address a review question, Google Scholar (www.scholar.google.es) was used as an additional source to search for articles on the metabolomic profile of camel urine. 

Following the premises given by Piasecki et al. [49], this freely accessible web search constitutes a great source of grey literature that is governmental alongside institutional reports that are sometimes not published in indexed journals, but which help to improve the conceptual comprehensiveness of qualitative systematic reviews. Lastly, seeking an in-depth review of the chemical information and experimental data identifying the biological activity of the small molecules present in camel urine, the freely accessible repository of PubChem (https://pubchem.ncbi.nlm.nih.gov) was used.

#### 2.1.2. Search Criteria

As search repositories are constantly updating their content, the evaluation of document databases was closed on October 1st, 2021. To filter among the vast number of documents that can be found in such repositories, the keywords ‘camel’, ‘urine’, ‘metabolomics’, ‘bioactivity’, ‘in vivo’, ‘in vitro’, and ‘effects’ were used in the ScienceDirect search engine without setting any limitations concerning time, date, language, and/or document type. The same search strategy was followed in Google Scholar to collect additional documents to be included in the database [50].

#### 2.1.3. Sample

A total number of 141 and 621 documents were collected from ScienceDirect and Google Scholar sites, respectively. From the first database, 140 references were related to camel urine’s bioactivity and one referenced camel urine metabolome. On the other hand, 614 and 7 references were related to camel urine’s bioactivity and the camel urine metabolome, respectively, in Google Scholar database. Only those documents in which either an in vitro, in vivo, or combined study of camel urine’s bioactivity was performed or those specifically dealing with the analysis and/or evaluation of the cytotoxic effects of active compounds of camel urine were retained in the database. Hence, those documents in which an anecdotal reference to urine, its bioactivity, and/or that of its compounds was made were considered to fall outside of the scope of this review and were thus discarded. Additionally, the final dataset comprised information on two reviews based on the bioactive effects of camel urine, while the rest of the documents described experimentally based studies. The first review included a detailed analysis of six already-published documents describing research studies on the therapeutic potential of camel urine, which were also individually considered [41]. Three of these papers had already been considered during the data collection process carried out when using the ScienceDirect repository [11,28,51]; hence, their duplicates were removed to ensure each document had been accounted for only once. The second review gathered a comprehensive theoretical presentation of the known traditional uses of some plants and animal-derived products and subproducts for cancer treatment in Algeria [52]. As a result, 132 and 614 documents were discarded from the databases initially constructed from ScienceDirect and Google Scholar, respectively.

As a result, final data sample consisted of 18 research papers (11 documents from ScienceDirect and 7 from Google Scholar database, wherein 10 were focused on camel urine’s bioactivity and 8 studied its metabolome).

### 2.2. Document Review

Data collected were sorted into two datasets: the first related to journal identification and impact indicators, and the second to the content of the documents. The information included in the first data set for each article was as follows: the journal in which the article was published or the conference proceedings in which conference contributions were presented, the year of publication/presentation, the yearly Journal Citation Report (JCR) impact factor of the journal in which the article was published, the total number of citations of each paper, the number of contributing authors, the country of the corresponding author, the camel species studied, the breeding location of the camels from which the urine was collected, and the number of animals comprising the sample, their sex, their average age, (in years) and their reproductive status. Each publication’s digital object identifier (doi) was also included in the database in order to be able to ensure document traceability and access each manuscript in case it was necessary a posteriori. 

Each journals’ JCR impact factor and total number of citations per paper was accessed from the Web of Science site. The computation of the impact factor for the journals publishing papers in 2021 (up to October) was executed by dividing the number of current year citations by the total number of items published during the 2 previous years, as suggested in the literature [53].

Afterwards, the second data set comprised the following information extracted from each article. For papers investigating the biological activity of the urine, special attention was paid to the different research methods and materials employed, focusing on the type of substrate used (raw or pre-processed urine), cell line(s) and/or organic tissue(s) tested, and model type (in vivo and/or in vitro). In studies analyzing the camel urine metabolome, the information retrieved consisted of the list of metabolites found, and the methodologies or protocols used for sample preparation, storage, and analysis. Then, for each reported chemical substance, the following information was registered: IUPAC name, synonyms, source(s) of origin/formation, and therapeutic and industrial uses. 

## 3. Results and Discussion

### 3.1. Bibliometrics Quantitative and Qualitative Analysis

Figure 1 depicts the outputs of the systematic document-screening stages. Afterwards, a description of the variables included in the present study, their type, and the levels that they correspond to is reported in Table 1. After testing the preliminary parametric assumptions (using the Shapiro–Francia test and Levene’s test to evaluate normality and homoscedasticity, respectively) (Appendix A) with the Stata software, version 15.0, the descriptive statistic parameters for normally and non-normally distributed data were calculated (Table 2) using SPSS software, Version 25.0 [54].

The reviewed documents on camel urine’s bioactivity (*n* = 12) and metabolome (*n* = 8) were published between 1996–2021 and 1925–2019, respectively. For these periods, the mean numbers of articles published per year for each topic were 0.48 and 0.08, respectively. Fourteen documents (six for urine’s bioactivity and twelve for the urine metabolome) were published under an open-access policy. Figure 2 presents the total number of articles published per topic and year. As it can be observed, large time gaps between the publication of papers are present for both research aims.

According to Faye [55], five levels of the economic importance of camel populations could be identified considering the percentage of Tropical Livestock Units (TLU) and density (number of camels/km^2^). First, those countries in which camels have a marginal importance (population represents less than 2% of the total TLU), such as South Asia and the Near-East, mainly, or anecdotal uses in Europe (Spain, The Netherlands, and France) (<1 camel/km^2^); countries in which they have a low economic importance (2–5% of the total TLU), such as Egypt, Libya, Central Asia, Iraq (<1 camel/km^2^), Pakistan, and Afghanistan (more than 2 camels/km²); countries in which camels have a medium importance (5–10% of the total TLU) such as Algeria, Kenya, and Ethiopia (<1 camel/km^2^); countries in which camels are quite important (10–25% of the total TLU), mainly Sahelian countries (1 camel/km²) and those in the Arabian Peninsula (more than 2 camels/km²); and those countries in which camels represented more than 25% of the whole TLU (Mauritania and Somalia) (more than 2 camels/km²). These two indicators (percentage of TLU and density) show that the economic importance of camels is quite predominant in Sub-Saharan countries and in the Arabian Peninsula.

As suggested by Iglesias Pastrana et al. [46], the relative economic and demographic importance of camels across countries translates into the greater or lesser attention paid by academicians and researchers to the study of the camel species. In this regards, in terms of internationalization, seven countries around the world have been involved in the research of camel urine’s bioactivity and/or its metabolomic profile, namely, Saudi Arabia (*n* = 13), Sudan (*n* = 2), and one paper per country in Algeria, China, Denmark, Canada, and Malaysia, respectively. The articles included in the present review were published in 17 different journals and one international conference. Most of the journals had an ethnopharmacology and biological chemistry background scope. The average number of authors per publication was five for camel urine’s bioactivity and three for its metabolomic profile. When the journal impact factor was considered, seven and three journals were indexed in JCR in the year in which the camel urine bioactivity and camel urine metabolome papers were published, respectively. 

The maximum JCR impact factor for the papers reviewed on camel urine’s bioactivity was 3.014, which was reached by an article focusing on camel urine’s inhibition of cytochrome P450 1a1 gene expression in murine hepatoma. This article was published in 2011 in the *Journal of Ethnopharmacology* [11]. On the other hand, the lowest JCR impact factor (0.692) was reached by an article published in 2019 in the *International Journal of Pharmacology*, which focused on the hepatoprotective effects of camel urine against carbon tetrachloride-induced liver toxicity [56] . 

Regarding the papers dealing with the camel urine metabolome, the maximum (3.138) and minimum (0.660) JCR impact factors were reached by two articles published in the *Saudi Journal of Biological Sciences* [57] and the *Indian Journal of Pharmaceutical Sciences* [58], respectively. Figure 2 illustrates the mean JCR impact index per topic reviewed and year. A parallel trend was described by the number of papers published and their academic impact.

The most highly cited paper on camel urine’s bioactivity was an article dealing with the cytotoxic effects of camel urine towards different human cancer cell lines, which was published in 2012 in the *Journal of Ethnopharmacology* [28], with a total of 47 citations. This value was close to that of the most highly cited article on camel urine metabolome, with 46 citations. It may be worth mentioning that the publication of the latter took place in 1925 in the *Journal of Biological Chemistry* [59] and constitutes the first simple analysis of some of the metabolites in camel urine. 

Three of the documents published in 2020, 2011, and 2016, which were considered in the present study, had not been cited at the moment in which the data collection carried out in the present study was finished, with respect to both camel urine’s bioactivity and metabolome-related literature. 

In summary, despite the time gap mentioned above, research efforts towards deepening the knowledge of camel urine’s bioactivity have been comparatively more regular in time and relevant in terms of the scientific impact of the publications that eventually were published than those focused on the metabolomic profile of this organic substrate. Indeed, although the first quantification of a discrete number of target metabolites in camel urine was published in 1925, no further research had followed until after almost one century. Indirectly, this provides evidence for the fact that there is a lack of a consistent time-overlap between both research topics. This, in turn, may have conditioned the interpretation and applicability of the results published, given the metabolomic profile and existing variability of the urines sampled had been barely known prior to bioactivity testing. Furthermore, a huge proportion of the papers reviewed were published in non-indexed journals, often resulting in a lack of peer-reviewed processes, which translated into poor scientific standards being applied. 

Regarding the number of authors involved in the studies, although the difference between camel urine’s bioactivity and metabolome is not large, the authors of the papers on camel urine’s bioactivity were more numerous. This finding could be explained on the basis that researchers of a wider range of disciplines are needed to properly investigate the bioactive effects of camel urine from a biomedical perspective. That is, specialists in molecular biology, physiology, biotechnology, and pharmaceutics usually congregate to contribute to the development of such studies. As it could be expected, the main authors of the literature reviewed are affiliated to academic institutions set in countries where camel production is well-established and the Muslim religion is practiced, i.e., where there is a local interest in valuing the camel urine. 

Relevant data are frequently missing, namely, regarding the total number of animals sampled (documents not reporting the information; *n* = 11), their mean age (documents not reporting the information; *n* = 13), physiological status (documents not reporting the information; *n* = 10), camel species (documents not reporting the information; *n* = 8), animal-breeding location (documents not reporting the information; *n* = 7), and sex (documents not reporting the information; *n* = 4). As a result, the potential variability in urine’s bioactive effects and metabolome, which could be ascribed to an animal’s intrinsic and extrinsic factors, cannot be determined and compared across studies. In general, adult virgin, pregnant, and lactating female dromedaries raised in Arabian countries are the most prevalent constituents of animal samples in the literature consulted for both topics, while males continue to be somehow avoided, possibly for socio-cultural preferences biased towards female urine [60]. Furthermore, there is a bias towards the use of dromedary camels rather than Bactrian camels, with just one report on the camel urine metabolome using urine from a female Bactrian camel [59]. 

Overall, the results of the bibliometric analysis illustrate that the sampling methodology is generally limited and the omission of information when publishing results in camel urine-focused research affects the interpretation, reproducibility, and replicability of scientific data [61].

### 3.2. The Bioactive Effects of Camel Urine: Current Status of Knowledge

Several studies conducted mainly during the last three decades have provided evidence for the anticancer, cardiovascular, gastroprotective, hepatoprotective, and antimicrobial effects of camel urine [13,24,27,31]. In this context, although the majority of the analyses have been performed in vitro, studies on living subjects also exist [41].

Among the papers considered, five studies tested the in vitro effects of camel urine toward murine [11] and human cell lines [28,62,63,64] and multidrug-resistant strains of *E. coli* [65]; four research works involved living mice [51,56,66,67]; one paper combined in vitro and in vivo experiments [68]; and a theoretical review reported the traditional use of camel urine for the treatment of cancer in a local Algerian population [52]. 

The biomedical application that was more frequently investigated was the potential use of camel urine for the treatment of oncological pathologies (*n* = 4). Still, other properties such as antiplatelet (*n* = 3), gastro- and hepatoprotective (*n* = 2), anticlastogenic (*n* = 2), and antimicrobial (*n* = 1) effects were also approached. These studies have shown that camel urine has cytotoxic effects against different human [28,52] and murine [11] cancer cell lines, in addition to presenting potential towards regulatory-related inflammatory angiogenesis [67]. Moreover, camel urine was reported to inhibit platelet aggregation [62,63,64], to protect against hepatic dysfunction [56], to prevent gastric ulceration [66], to act as an anti-clastogenic factor [51,68], and to aid the treatment of certain infectious diseases [65].

All the aforementioned studies used whole camel urine as their bioactive substrate, with differences being mainly ascribed to sample processing: raw unprocessed urine (*n* = 9), lyophilized urine resuspended in PBS immediately before utilization (*n* = 2), and fresh sterilized urine mixed with distilled water (*n* = 1). 

Anecdotally, the bioactivity of other substrates was also referenced or tested in literature. For instance, Al-Yousef et al. [28] cited that some patients drink camel urine mixed with camel milk as an unconventional chemotherapeutic regimen for cancer treatment in the Arabian Peninsula, which would be later supported by Gupta et al. [69] who recently demonstrated the anticancer properties of this traditional practice. Additionally, the bioactive fraction PMF (Prophet Medicine Fraction) and subfraction PMF-K, obtained by fractioning previously lyophilized camel urine (PM701), have been highlighted as selective anticancer [70,71] and antimicrobial [72] agents. In line with this, the registered patents for these pharmaceutical formulations state that the fractionation of the total urine will strengthen the efficacy of this animal-derived substance but also its preparation into capsules or combination with nanoparticles so as to help target the cancer tissues without harming the normal ones [73,74]. In this regard, the authors also discuss and address the implications of formulation in the enhancement or deterioration of camel urine’s bioactivity. Similarly, cow urine has been granted US Patents for its antibiotic, antifungal, and anticancer properties [75], and human urine is processed to obtain a peptide fraction that is used in the treatment of allergic diseases and autoimmune processes [76]. In general, the acknowledged bioactive effects of camel urine are hypothetically ascribed to the bioactive compounds present in the desert plants upon which these animals feed [24,77,78]. In parallel, the research postulates that the single-domain antibodies found in camel blood, which present rapid renal clearance, are the main responsible elements for the presumable bioactivity of camel urine [79,80]. 

Still, these hypotheses need to be further tested through the active work of research groups on camel urine’s bioactivity and metabolome. Among other factors, specific attention must be paid to record the information related to the breeding location, production system, and diet of the animals whose urine is intended to be tested for bioactivity and/or characterized for its metabolomes, which is supported by the preliminary conclusions drawn by Ali et al. [81] and Elkhair [82], who reported that the free or restricted access to feed and water as well as the nutritional status of the animals may more greatly influence renal function and urine excretion than other factors such as age. 

Another methodology-related bias that may extensively affect research outcomes is the lack of knowledge regarding the osmolarity of the camel urine before experimental testing. If we consider the physiological adaptation of camels to desert conditions, their renal physiology is well known to function in favor of water retention and urine concentration. Hence, when highly concentrated urines are tested for bioactivity, toxicity may arise from the osmolarity values that exceed the cells’ tolerance limits [83,84], and not from specific urine’s bioactive compounds. Hence, given that the tolerability range to different osmolarity conditions is expected to vary across cell lines [85], it can be crucially important to measure urine’s osmolarity before experimental testing and to pre-process the samples as needed to adapt the osmotic concentration to the specific cell(s) line(s) under study. Contextually, the evaluation of camel urine prior to studying its properties and inner characteristics such its osmolarity, together with a deeper knowledge of the metabolomic profile, may help researchers study the substrates that better suit the subsequent aims of research and may enhance the development of strategies aimed at functionally valorizing the use of camel urine as an alternative source of potential drugs’ discovery and isolation.

### 3.3. Camel Urine Metabolome: A Detailed Overview and Future Prospects for Biomedical Research

Once the biological activities of whole camel urine are tested and confirmed under the experimental assumptions described in the previous subsection, the following step is the understanding of the relationship between the chemical composition and bioactivity patterns of each urine tested. 

Although the use of web-based platforms for the prediction of bioactivity and the associated beneficial effects of chemicals can support the process of new drugs’ discovery, it is highly recommended to additionally perform evaluations of key biochemical parameters (i.e., permeability) and design functional assays (in vitro/in vivo) given the intrinsic limitations of prediction rates that can sometimes arise from in silico results when approached separately [32,86].

Table 3 lists the metabolites whose presence in camel urine has been reported to date. Generally, urine samples were extracted with organic solvents [56,57,87]. After derivatization with a silylating reagent, the analytes were identified through GC-MS analysis by comparison with the spectra included in the National Institute of Standard and Technology (NIST) [57,88] based on a comparison of the unknown analytes’ mass spectrum’s peaks to those of the peaks in the library’s spectra (Match factor > 800) [89]. Careful inspection of the analytical methods used (generally roughly described in most of the cited papers) compelled the exclusion of several compounds that were “found” in urine. which were likely food contaminants, artifactual derivatives, or compounds coming from the partial degradation of the stationary phases (Si-containing compounds). Only organic compounds were reported, and the heavy metal complexes reported in the original papers as “found” were excluded.

As expected, many of the enlisted metabolites are relatively hydrophilic with their corresponding calculated partition coefficient (cLog*P*) values often being ≤ 0 (Figure 3, panel A; Appendix A). This should prevent any systemic pharmacological effect for most of them since cLog*P* values of pharmacologically relevant compounds generally fall within the 0 to 3 cLog*P* range [90]. This inference is in agreement with the very low drug-likeness shown by most of the major constituents in camel urine (cf. Appendix A and panels B and C in Figure 3; Oprea [90], Ghose et al. [91]). A fraction ≥ 75% of the stated metabolites should be endowed with good oral bioavailability since they do not violate the ‘Rule of Five’ (Ro5) test [92] and Veber’s filter for oral absorption [93] (Appendix A and panels D and E in Figure 3). The major components found in camel urine are small molecules (MW = 200 ± 100; Appendix A) and 38% of them could be considered as useful fragments to develop rather complex derivatives with good expectations for biological activity since they fulfil the ‘Rule of Three’ (Ro3) proposed for fragment-based lead discovery (Congreve et al. [94]; panel F in Figure 3). The quality of those fragments is high in most cases as may be inferred from the high fraction of sp^3^-hybridized carbon atoms (Fsp^3^) found in most of them and the frequent presence of chirality centers. High Fsp^3^ and chirality are positively related to the rate of success in clinical development (Panels G and H in Figure 3, and Appendix A; Lovering et al. [95]).

**Table 3 ijms-23-15024-t003:** List of major metabolites reported to be present in camel urine.

IUPAC Name	References
2-Methylbutanedioic acid	[56,57,96]
Propanedioic acid	[56,57]
*2-Aminopropanedioic acid*	[56,57,96]
*(2S,3R)-Butane-1,2,3,4-tetrol*	[56,57,96]
*(2S)-2-Amino-4-(diaminomethylideneamino)oxybutanoic acid*	[56,57]
*2-Amino-3-methyl-4H-imidazol-5-one*	[56,57,59]
*(3R,4S,5R,6R)-6-(Hydroxymethyl)oxane-2,3,4,5-tetrol*	[56,57]
*(2S,4R)-Pentane-1,2,3,4,5-pentol*	[56,57]
*Nonanedioic acid*	[56,57,89]
2-Benzamidoacetic acid	[56,57,59,96]
2-(*N*-Acetylanilino)acetate *	[56,57]
(2*R*)-2-[(2*S*,3*R*,4*S*)-3,4-Dihydroxy-5-oxooxolan-2-yl]-2-Hydroxyacetaldehyde	[56,57]
Hexadecanoic acid	[56,57]
3-Phenylpropanoic acid	[56,57]
*7-[3,5-Dihydroxy-2-(3-hydroxyoct-1-enyl)cyclopentyl]heptanoic acid*	[56,57]
5-[(2*S*,3*R*,4*S*,5*R*)-3,4-Dihydroxy-5-(hydroxymethyl)oxolan-2-yl]-1*H*-pyrimidine-2,4-dione	[56,57,96]
(3*R*,4*S*,5*S*,6*R*)-6-[[(2*S*,3*R*,4*S*,5*R*,6*R*)-3,4,5-Trihydroxy-6-(hydroxymethyl)oxan-2-yl]oxymethyl]oxane-2,3,4,5-tetrol	[56,57]
(4*R*,5*R*,6*R*)-6-(Hydroxymethyl)oxane-2,4,5-triol	[56,57]
(*E*)-Octadec-9-enoic acid	[56,57,89]
2-Hydroxypropanoic acid	[96]
Acetic acid	[88,89,96]
(2*S*)-2-Aminopropanoic acid	[96]
2-Aminoacetic acid	[96]
Oxalic acid	[96]
2-Methylphenol;3-methylphenol;4-methylphenol	[96]
2-Hydroxy-2-methylpropanoic acid	[96]
3-Hydroxy-3-methylbutanoic acid	[96]
Urea	[59,96]
Benzoic acid	[88,89,96]
*2-Phenylacetic acid*	[96]
Benzene-1,2-diol	[96]
*2-Hydroxybenzoic acid*	[89,96]
3-Methylhexanedioic acid	[96]
1-Methylimidazol-2-amine	[96]
3-Hydroxybenzoic acid	[96]
3-Hydroxy-3-(3-hydroxyphenyl)propanoic acid	[96]
Heptanedioic acid	[96]
*2-[(2-Hydroxybenzoyl)amino]acetic acid*	[96]
7,9-Dihydro-3*H*-purine-2,6,8-trione	[96]
2-[(3-Hydroxybenzoyl)amino]acetatic acid **	[96]
(3*S*,4*R*,5*S*)-5-[(1*R*)-1,2-Dihydroxyethyl]oxolane-2,3,4-triol	[96]
(2*S*,3*S*,4*S*,5*R*,6*S*)-3,4,5-trihydroxy-6-(4-methylphenoxy)oxane-2-carboxylic acid	[89,96]
(2*S*,3*R*,4*S*,5*R*)-3,4,5,6-Tetrahydroxyoxane-2-carboxylic acid	[96]
3-Methylheptan-4-one	[88,89]
Butyl butanoate	[88,89]
1-*N*,1-*N*,2-*N*,2-*N*-Tetrafluoro-2-methylpropane-1,2-diamine	[88]
1,1-Dibutoxybutane	[88,89]
Pentanoic acid	[88]
Butyl 4-hydroxybenzoate	[88]
Hydroxylamine	[88]
*(9Z,12Z,15Z)-octadeca-9,12,15-trienoic acid*	[88]
Creatine	[59]
9-Methylanthracene	[87]
1-Methyl-7-propan-2-ylphenanthrene	[87]
5-Methyl-6-phenylpyrazine-2,3-dicarbonitrile	[87]
1,2-Dichloro-4-ethylbenzene	[87]
*6,15-Dimethyltricyclo [10.4.0.04,9]hexadeca-1(12),4(9),5,7,13,15-hexaene*	[87]
2,3,5-Trimethylphenanthrene	[87]
*1-(2-Hydroxyphenyl)-3-phenylpropane-1,3-dione*	[87]
1,1-Diphenylprop-1-ene-2-thiol	[87]
2,5-Dimethyl-4-oxidopyrazin-1-ium 1-oxide	[87]
1-Isothiocyanato-2-methylsulfanylbenzene	[87]
Benzo[*f*][1]benzothiole	[87]
4-Methyldibenzothiophene	[87]
(3*S*,3a*S*,5a*S*,9b*S*)-7-Chloro-3,5a,9-trimethyl-3a,4,5,9b-tetrahydro-3*H*-benzo[*g*][1]benzofuran-2,8-dione	[87]
*1-Methyl-3,7-dihydropurine-2,6-dione*	[87]
Bicyclo [4.2.0]octa-1,3,5-triene	[87]
9-Azatricyclo [10.4.0.02,7]hexadeca-1(16),2,4,6,12,14-hexaene	[87]
1,2,3,4,6,7,8,11,12,12b-Decahydrobenzo[*a*]anthracene	[87]
Phenol	[89]
*(E)-3-phenylprop-2-enoic acid*	[89]
Butyl hexadecanoate	[89]

Compounds in italics are known to have bioactive properties with applications in human health and therapeutics. * As its trimethylsilyl ester; ** Found as its bis-trimethylsilyl derivative.

In total, seventy-two compounds detected in camel urine were assumed to be highly abundant. Out of these, twenty-seven were found in at least two of the studies examined. This indicates that these compounds may be present across different urine samples regardless of the intrinsic and extrinsic (dietary and environmental, among others) factors that vary between animals and across species, and thus probably arising from common metabolic pathways.

The evaluation of the urinary metabolomic profile of other species is scarce, specifically if we aim at finding resources in which the bioactive role of urine metabolites is evaluated. Among those twenty-seven metabolites common to camel urine metabolomics studies, seven bioactive compounds are shared with cattle (propanedioic acid, hippuric acid, butanedioic acid, ribitol, D-glucuronic acid, hexadecenoic acid, and prostaglandin f1α) [86], two with goats (hippuric acid and glycine) [97], one with sheep (creatinine) [98], and four with giraffes (methylsuccinate, benzoate, hippuric acid, and creatinine) [99].

On the other hand, the fact that certain chemicals could only be exclusively found in one study may reflect the potential variability in urine composition and bioactive effects across different camel subjects, breeds, and populations for animal-related factors influencing metabolism such as sex [100,101], age [102], diet [97,103], environmental seasonality [104,105], composition of gut microbiome [106,107], and the kinetics of the reactions leading to the production of metabolites or the time course of metabolism [108]. For these reasons, it is strongly recommended that details on the animals’ sex, age, physiological status, rearing conditions (desert-living vs. farming), and diet formulation/composition are provided alongside the animal subjects or groups from which the urine was collected.

Moreover, the compounds detected depend on the analytical strategy employed, from sample preparation to the technique used for compositional profiling [109,110,111]. In the metabolomics studies reviewed, the sample preparation methods included liquid–liquid extraction with organic solvents, such as dichloromethane [65,86], diethyl ether [57], chloroform, and ethanol [96], or solid phase extraction [58]. Additionally, some studies employed enzymatic treatments to remove urea [95] or to hydrolyze glucuronides and sulphate-conjugated compounds [58]. Naturally, the conditions used in the extraction procedure will significantly affect the obtained compounds. For instance, Ahamad, Alhaider, Raish, and Shakeel [86] optimized the extraction solvent and identified several carboxylic acids from which hippuric acid appears in higher concentrations. It should be highlighted that the authors did not perform a quantitative analysis. On the other hand, El-Nadi and Al-Torki [57] used a less polar solvent and naturally identified less polar compounds, mainly aromatic derivatives. Additionally, knowing that some more abundant compounds can hinder the observation of other metabolites, some authors employed enzymatic treatments to remove urea [95] or to hydrolyze glucuronides and sulphate-conjugated compounds [58]. One of those examples is the work of Khedr and Khorshid [58], where the authors were able to quantify some phenolic compounds, including interesting bioactive compounds such as salicylic and cinnamic acids, in camel urine after a pre-treatment with β-glucuronidase aryl-sulphatase.

Regarding the analytical platform used, all studies employed gas chromatography coupled with mass spectrometry (GC-MS). Some studies analyzed only volatile compounds [82,89], whereas others included a derivatization step, mostly silylation, to detect less volatile metabolites such as amino acids and phenolic compounds [58,86,95]. An important limitation of these works is that they did not report accurate quantification data. Indeed, as far as we could perceive, none used the internal standard method, which is the most recommended approach in MS-based analysis to compensate for variable response factors arising from matrix effects.

As mentioned before, sample derivatization prior to GC-MS enables the range of detected compounds to be enlarged. However, it often implies the use of lengthy procedures, which may hinder reproducibility and introduce artifacts that complicate data analysis. Techniques such as LC-MS and NMR spectroscopy represent attractive alternatives for analyzing the urinary non-volatile metabolic profile. To our knowledge, their application to the study of camel urine has not been reported yet, although they have been widely employed in the analysis of human urine, as recently reviewed elsewhere [112,113]. Benefiting from important advances in separation technologies, such as HILIC (hydrophilic chromatographic separation), which allows for the improved separation of highly polar compounds, and UPLC (ultra-performance LC), which enables significant improvements in sensitivity, LC-MS offers great advantages for urinary analysis. As for NMR, although it has lower sensitivity (sub-mM) than MS-based methods (<pM) (and thus a lower metabolome coverage), its associated simple sample preparation protocols, non-destructive nature, unparalleled reproducibility, and inherently quantitative power represent attractive features that justify its continued application in the metabolic profiling of complex biofluids such as urine. Hence, future applications of LC-MS and/or NMR profiling to the study of camel urine are expected to significantly complement the GC-MS studies already reported and to disclose significant new information on urinary metabolites.

In this context, the standardization of the techniques used for the study of the metabolomic profile of camel urine, linked to the increase in the sample variability, would allow for the statistical analysis of the potential influence of the different abovementioned animal-related factors on the excreted metabolites. Concerning diet, given that the majority of the studies that report a significant effect of diet on the metabolites excreted are performed under laboratory conditions with animals fed with relatively standardized diets, it can be expected that there will be pronounced differences found regarding the kind and proportion of metabolites excreted in animals that are allowed to freely graze and thus have more variable diets [107], such as camels. Similar to the recently published review on camel milk’s bioactive peptides [114], we explored the literature on the main sources and biological effects for each chemical found in camel urine (Appendix A). The most-reported compounds are plant structural compounds or metabolites and products of bacterial metabolism. Their main applications can be summarized under the following categories: plastic fabrics, cosmetics, food additives/preservatives, anticancer agents, analgesic/anti-inflammatory compounds, and biocides. In general, the most commonly reported uses can be classified within the first three categories. Nonetheless, more in vivo evidence is needed to further recognize the potential industrial/pharmacological uses of all these chemicals, together with a wider knowledge of the possible interactions/interferences between urine compounds.

To the best of our knowledge, the metabolomic profiling of camel urine followed by bioactive testing has only been performed in one study, which addressed the protective role of camel urine against carbon tetrachloride-induced liver toxicity [65]. Instead, the seven additional papers considered were isolated evaluations of the metabolomic profile of camel urine, without a posterior evaluation of its potential bioactive effects. It is also interesting to note that, as for urine bioactivity-related research, a sex bias is widely present in metabolomic studies, since the majority (*n* = 6) of these studies characterized female camel urine, and the rest (*n* = 2) did not indicate the sex of the sampled animals. 

Furthermore, taking into consideration the reported absence of genetic toxicity related to the intake of camel urine [67,115,116], its low content of urea and ammonia (the main compounds responsible for the characteristic odor of the biofluid), and its rich mineral salt composition and higher concentration in comparison with the urine from other mammals [28], the exploration of the unique chemical constituents of camel urine and their different molecular mechanisms of action against different pathological conditions should be considered as an emerging strategy deserving further study concerning the assessment of camel urine exploitation as a natural source for the discovery and isolation of new chemicals with valuable pharmaceutical potential.

The motivation of the present research is not to promote the consumption of raw camel urine, but to value its role as a source for specific molecules with bioactive potential, which can also be safety-evaluated with respect to medicinal chemistry and drug discovery in a public health scenario. Indeed, the Shia code explicitly asks for caution in the consumption of urine from animals. According to Islamic laws in reference to the use of medicines from halal sources, “it is haraam to drink the urine of all haraam animals, and also of those whose meat is halal to eat, including, as an obligatory precaution, that of a camel. However, the urine of a camel, a cow or a sheep can be consumed, if recommended for any medical treatment” [117,118].

This is supported by the fact that camel urine has been compared to other drugs since more than a decade ago. Contextually, Kabarity et al. [119] reported camel urine’s cytotoxic effect to be comparable with that of cyclophosphamide, a standard drug used in cancer chemotherapy. However, these and other authors reported the absence of a significant clastogenic effect from consuming camel urine, namely, a mitodepressive effect. Indeed, as suggested before by Anwar, Ansari, Alamri, Alamri, Alqarni, Alghamdi, Wagih, Ahmad, and Rengasamy [67] the lack of such a clastogenic effect may be dose-dependent in light of their finding of 25 and 50 mL/kg of camel urine treatment significantly improving the cyclophosphamide-induced clastogenic effect, while the mitodepressive effect may necessitate a rather reduced concentration.

Additionally, regarding the protective effects of camel urine against peptic ulcers, Salamt et al. [41] suggested that a treatment of 5 mL/kg of camel urine results in 100% ulcer inhibition in HCl/EtOH and WRS models, which was reduced to 66.7% in an indomethacin model. Furthermore, there was a 100% healing rate (no ulcers observed) in indomethacin-induced gastric damage (healing model) compared with cimetidine, which only resulted in a 60.5% healing rate. However, the doses of camel urine currently suggested as effective cannot be accurate since the methodological approaches have been a single run instead of an iterative process (five steps or phases), which is the accepted standardized strategy for drug development in medicinal chemistry.

Thus, despite the aforementioned assessment of the bioactivity and putative beneficial effects of camel urine towards oncological, cardiovascular, digestive, and infectious pathologies, substantial knowledge gaps have been identified within the scientific literature on the subject. The reduced sample size and high variability, along with the non-indication of specific intrinsic and extrinsic factors and the disregard for urine’s properties such as its osmolarity, could lead to the misinterpretation of the bioactive effects of camel urine, hence leading researchers to formulate biased conclusions. Furthermore, as the metabolome of this animal-derived secretion is hardly determined before bioactivity assays, potentially bioactive small molecules cannot be proposed and used for validation tests. Among other factors, the preliminary physical–chemical status of the urines when used for bioactivity or metabolomics analysis (i.e., lyophilized) should be further studied and considered as an influencing factor given its potential conditioning effects on research outputs. Consequently, future research initiatives should focus on disentangling the multi-etiological background of the camel urine metabolome and its specific potential implications for public health promotion through preclinical studies aimed at testing the bioactivity of the individual molecules contained in the urine of these animals but that, in any case, do not promote the consumption of this biological fluid in a raw status. Different approaches may be useful in this respect, including tandem metabolite profiling and bioactivity screening of whole urines or their selected fractions to identify bioactive candidates. Ideally, this should then be followed by assessments of individual compounds to validate their putative biological effects in vitro and in vivo. This may allow for the determination of dose-dependent effects, safety levels, and the potential inclusion of specific metabolites in the formulation of new drugs.

## 4. Conclusions

Although the research addressing camel urine has existed for more than a century, large time gaps have occurred until the present. The clear lack of connection between metabolomics strategies and camel urine’s composition and bioactivity has translated into the limited knowledge of this important biological matrix. Most of the scarce articles published in this context have emerged from few countries and have been published in journals with ethnopharmacology and biological chemistry scopes. Still, the importance of the subject has been duly recognized; as such, the publications have tended to reach high impact levels. According to the current knowledge, a considerably larger number of bioactive metabolites can be found in camel urine compared to that of other affine species. This explains the patent interest in the topic within the scientific community, especially in fields related to potential human benefits or drawbacks derived from the use of camel urine (e.g., anticancer, cardiovascular, gastroprotective, hepatoprotective, and antimicrobial effects, among others). Among the main limitations of the research work developed to date, sampling-related problems and the omission of information have been identified as significantly affecting the interpretation, reproducibility, and replicability of data. In addition, a larger number of in vivo studies may need to be implemented in contrast to in vitro studies. Particularly, osmolarity and alkalinity parameters need to be controlled prior to the undertaking of research aiming at testing bioactivity. All in all, enhanced multidisciplinary, comprehensive studies are required to evaluate the implication of intrinsic and extrinsic factors in camel urine’s composition and properties, thus enabling the determination of the conditions and levels at which bioactivity is developed to ensure future reliable applications of camel urine’s components in biomedicine.

## Figures and Tables

**Figure 1 ijms-23-15024-f001:**
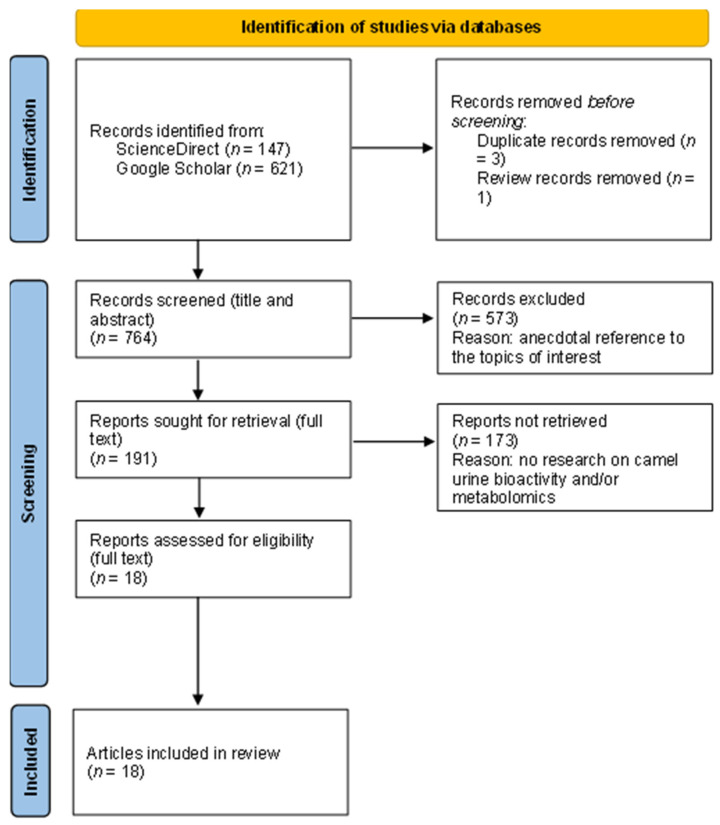
Process flow chart showing the sequential steps of literature search and review in the present study.

**Figure 2 ijms-23-15024-f002:**
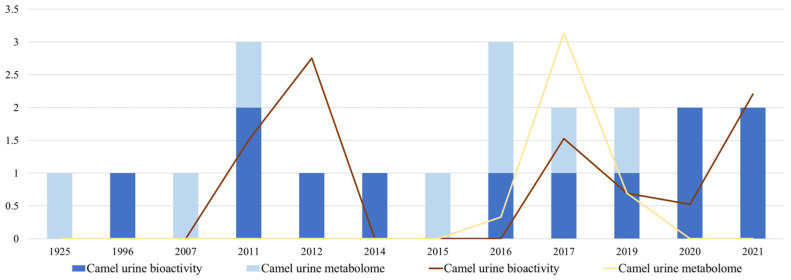
Number and impact factor of camel urine- and metabolome-related publications from 1925 to 2021. Bars represent the number of documents published per topic and year, while straight lines depict each year’s respective mean JCR impact index.

**Figure 3 ijms-23-15024-f003:**
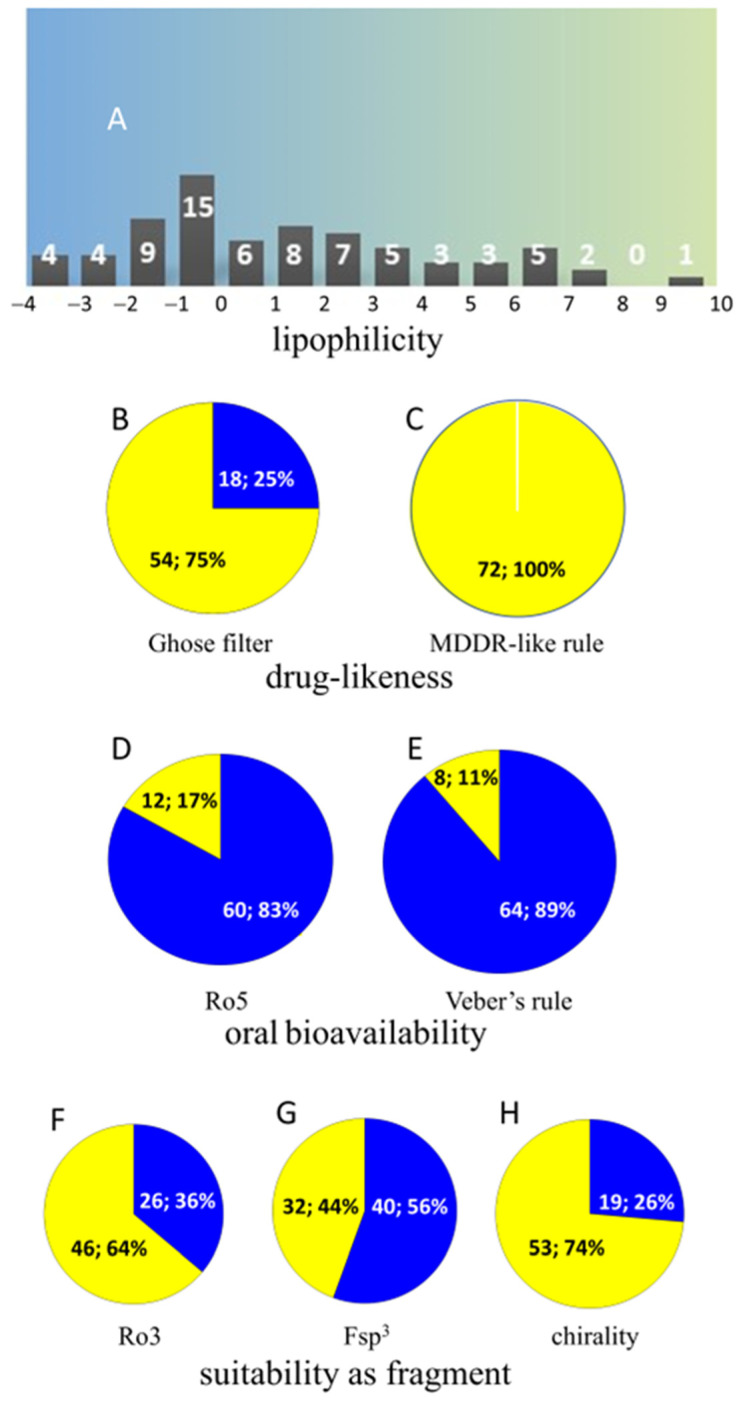
A: distribution of cLog*P* (ACD/Labs 2020.1.0) in camel urine metabolome; B: number and percentage of camel urine chemicals violating the drug-like Ghose filter [−0.4 < cLog*P* < 5.6, 160 < MW < 480, 40 < molar refractivity [89] < 130, 20 < number of atoms < 70 [90], yellow sector]; C: number and percentage of camel urine chemicals violating the MACCS-II Drug Data Report (MDDR)-like rule [ring number (RN) ≥ 3, rigid bonds (RBs) ≥ 6, RBs ≥ 18 [89]] for drug-like compounds (yellow area); D: number and percentage of camel urine chemicals fulfilling the Rule of Five’s (Ro5) requirements [hydrogen bond donors (HBDs) ≤ 5, hydrogen bond acceptors (HBAs) ≤ 10, MW ≤ 500 Da, cLog*P* ≤ 5 [91]] for oral bioavailability (blue sector); E: number and percentage of camel urine chemicals fulfilling Veber’s rule [polar surface area (PSA) ≤ 140 Å2, rotatable bonds (RBs)≤ 10, HBDs + HBAs ≤ 12 [92]] for oral bioavailability (blue sector); F: number and percentage of camel urine chemicals fulfilling the Rule of Three’s (Ro3) requirements [hydrogen bond donors (HBDs) ≤ 3, hydrogen bond acceptors (HBAs) ≤ 3, MW < 300 Da, cLog*P* ≤ 3 [93]] for fragments (blue sector); G: number and percentage of camel urine compounds with a fraction of sp3-hybridized carbon atoms (Fsp3) > 0.2 [94] (blue sector); H: number and percentage of chiral compounds found in camel urine.

**Table 1 ijms-23-15024-t001:** Category description for bibliometric variables registered.

Variable	Type	Levels	
		Camel Urine’s Bioactivity	Camel Urine Metabolome
Journal	Nominal	10 Scientific Journals	7 Scientific Journals and 1 International Conference Paper
Year of publication	Ordinal	1996 to 2021	1925 to 2019
JCR Impact Factor per paper publication year	Numeric	0 to 3.014	0 to 3.138
Total number of citations per paper	Numeric	0 to 47	0 to 46
Number of authors	Numeric	1 to 12	1 to 4
Country of corresponding author	Nominal	Algeria, Canada, Malaysia, and Saudi Arabia	China, Denmark, Arabia Saudi Arabia, and Sudan
Camel species	Nominal	*Camelus dromedarius* and Not indicated ^1^	*Camelus dromedarius*, *Camelus bactrianus*, and Not indicated ^1^
Camels’ breeding location	Nominal	Algeria, Egypt, Saudi Arabia, Somaliland, and Not indicated ^1^	China, Egypt, Saudi Arabia, and Not indicated ^1^
Sample size	Numeric	3 to 67 (Not indicated in 7 documents) ^2^	1 to 23 (Not indicated in 4 documents ^2^)
Sex of sampled animals	Nominal	Male, female, and Not indicated ^1^	Female and Not indicated ^1^
Mean age of sampled animals (years)	Numeric	3.5 to 6 (Not indicated in 7 documents ^2^)	2.5 to 6 (Not indicated in 6 documents ^2^)
Physiological status of sampled animals	Nominal	Lactating females, Physiological status cluster 1 (virgin, pregnant, and lactating females), and Not indicated ^1^	Lactating females, pregnant females, Physiological status cluster 2 (virgin and lactating females), and Not indicated ^1^

^1^ Qualitative data not detailed; ^2^ Quantitative data not detailed. When testing the statistics’ parametric preliminary assumptions, ‘Not indicated’ data for nominal variables were considered as a different category, while ‘Not indicated’ data for ordinal and numeric variables were treated as missing values.

**Table 2 ijms-23-15024-t002:** Summary statistics for the ordinal and numeric variables.

**Camel** **Urine’s Bioactivity**			
**Normally distributed variable**	**Mean**	**Standard deviation**	**Min/Max**
JCR Impact Factor per paper publication year	1.2	1.1	0/3.014
Number of authors	5	3	1/12
Sample size	25	27	3/67
**Non-normally distributed variable**	**Median**	**Mode**	**Interquartile range**
Year of publication	2016	2011	25.00
Total number of citations per paper	7.0	1.0	3.0
Mean age of sampled animals (years)	6.0	6.0	2.5
**Camel** **urine metabolome**			
**Normally distributed variable**	**Mean**	**Standard deviation**	**Min/Max**
Number of authors	2.9	1.1	1/4
**Non-normally distributed variable**	**Median**	**Mode**	**Interquartile range**
Year of publication	2015	2016	94.00
JCR Impact Factor per paper publication year	0.0	0.0	3.1
Total number of citations per paper	3.0	0.0	46.0
Sample size	1.0	1.0	22.0
Mean age of sampled animals (years)	4.2	2.5	3.5

## Data Availability

The data presented in this study are available on request from the corresponding author.

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
