# Peer review of "Camel (Camelus spp.) Urine Bioactivity and Metabolome: A Systematic Review of Knowledge Gaps, Advances, and Directions for Future Research"

_ijms, 2022, doi:10.3390/ijms232315024_

Round 1
Reviewer 1 Report
Reviewer’s comments on manuscript
Camel (Camelus spp.) urine bioactivity and metabolome: knowledge gaps, advances and directions for future research
I have carefully read the MS and checked the references stated in the text.
Even though at first glance the review looks as it is very well organized, after careful evaluation, there is no medical/therapeutic logic between the information in the text and the supposed uses of this urine.
I raise the following questions and remarks:
1. In the abstract the authors talk about “studies on the therapeutic properties of camel urine and the detailed metabolomics characterization of its chemical composition are scarce and often unrelated”. What is the scientifically proved medical justification of the use of a waste product of mammalian metabolism to treat humans in the first place?
2. The statement “camel urine intra and interindividual variability in terms of chemical composition” is not explained scientifically with the animals’ metabolism, but the authors rely solely on differences between the analytical techniques used. This is a general misunderstanding of physiology.
3. in Introduction:
a) lines 40-45: Why the references used to “justify” the statement are historical interpretations of Mesopotamian texts? In these references, it is clearly stated that “it is believed” and “it is assumed that…” I think these publications are not a source of proven medical knowledge. Refs. 2 and 3 are even not in English and quite hard to obtain.
b) Lines 48-57 are another historical interpretation, which is neither medically sound, nor came from a medical publication, but rather philological one. How the authors rely on this?
c) Line 61: even cancer is a leading health concern, autourotherapy has neither been studied, nor practiced commonly. How do you explain this isolated cultural phenomenon from the viewpoint of evidence-based medicine?
d) lines 66-68: There is a chemical misinterpretation of the statement. The potassium and magnesium ions are not directly connected to alkaline pH, but are a result of the renal filtration in alkaline urine. Another misconception here is the statement that albuminuria is a normal condition. If albumin in urine would have pharmacological effect, it should be in levels, high enough to be a diagnostic criteria of albuminuria, thus kidney damage. This is not discussed. How do you justify your views?
e) lines 72 to 81 deals with a publication of in vitro cytotoxic effect of urine. This is only one publication, from which it is clearly visible that the cytotoxicity described is both osmotic, from the concentrations used, and due to the toxic metabolites, which the urine contains as a waste product. In vivo results are neither found in literature, nor could in vivo cytotoxicity be extrapolated in an animal. This misinterpretation is a general lack of understanding the difference between cell models and animal models to test activity.
f) lines 82-93: Why there is no information, in this section, about the toxicity of metabolites found in urine. It is known from physiology, that if a compound is “precious” for a mammalian organism, it is reabsorbed in the nephron. How do you justify physiologically that “active compounds” are excreted? Moreover, why there is no information on the cytotoxicity of the metabolites as isolated compounds?
g) lines 98-104: even though some analytical methods are reported in the MS, no critical review was given on them, nor there is any kind of detailed chemical information on neither the capabilities of the methods, nor the findings in urine.
4. Review methodology: this is another issue of the MS – why, even though the authors clearly state that a respected search engine as Science Direct gives only one publication on camel urine, results from Google were used as well? The justification that
“governmental and institutional reports that are sometimes not published in indexed journals, but which helps to improve the conceptual comprehensiveness of qualitative systematic reviews.” is not scientifically sound. It rather looks as a conspiracy theory. If a study is good, correctly conducted and the results are sound, there is no problem of publishing it in a respectable scientific journal. A general problem with the Internet is that it is full of unproven information. The MS clearly lack the critical conception of that in mind.
5. Lines 173-177: the journal’s IF is not directly connected to medical soundness of the matter. How do you explain that supposed connection in your text?
6. Table 1: Why there is no statement how many papers, dealing with camel urine, from a medical viewpoint, the authors have found in the first place? For a 96 years’ period if this urine is really effective, it should have been at least several. Why there was a necessity to elongate the search terminology, if the camel urine is so widely used?
7. How do you explain “not indicated” in Table 1? This concerns the species of the camels, the breeding location of the animals, the sample size, their sex and their age. These are variables that in general are responsible for statistical differences between animals and their urine. If you accept articles, in which those are not stated, what is the statistical significance of your review in the first place?
8. lines 224 to 237 are a general talk on camels’ importance as a livestock and have nothing in connection with neither the topic of the MS, nor with the scope of the Journal.
9. lines 263-260: How do you explain your views, taking in mind my remark of the same (see 3e, above)?
10. Paragraph 3.2: This is a statistical interpretation of publications found. It is neither an evidence-based, nor statistically sound connection between the effect stated and the sample size. How do you justify statistically results, accumulated in less than five papers, some of which even written in non-medical and non-chemically oriented journals?
11. There is no critical interpretation of lines 498 to 502. Neither the oral dose of urine is justified in the reference stated, nor is the water immersion-restraint stress (WRS) model an approved and fully working model.
12. Why there is no information on the biochemical possibility of some organic compounds to be found in the urine, or their eventual toxicity in Table 3? Even though compound as thiophene, pyridine, quinolone and even phenol are presented as “found’ in urine, there is no critical review neither of this possibility, nor of the possible health damage that they would induce in an organism after ingestion. Moreover, the statement that Acetic acid was found in urine, even though the authors clearly state that camel urine is alkaline is a very strange understanding of chemistry.
I strongly recommend rejection.
Author Response
Reviewer 1
I have carefully read the MS and checked the references stated in the text.
Even though at first glance the review looks as it is very well organized, after careful evaluation, there is no medical/therapeutic logic between the information in the text and the supposed uses of this urine.
Response: The traditional therapeutic benefits of camel urine can be collectively considered as the treatment of cancer and certain infectious, gastric and cardiovascular abnormalities and diseases (Salamt, Idrus, Kashim, & Mokhtar, 2021). Some scientists have reported that the medicinal properties of both camel’s milk and urine ascribe to the gamma globulins and other immune components, such as immunoglobulins, which can be found in both biological substrates (A. Alhaider et al., 2013; A. A. Alhaider, Bayoumy, Argo, Gader, & Stead, 2012).
Fifty percent of the circulating antibodies in camel blood are built combining two heavy chains while they lack light chains. This in turn makes them them one-tenth the size of human antibodies. This smaller size provides them with the advantage of being transmitted through the milk of the lactating camel, but also permits their cross through the blood-brain barrier or their filtration via urine. In addition, the small size makes them available to be absorbed from the gut into the general circulation of individuals who have consumed camel’s milk and/or urine (Hamers-Casterman et al., 1993). The single antigen-binding domains (VHH) of these heavy-chain antibodies, also known as nanobodies, have been used in the diagnosis and treatment of cancer, as well as in the development of biosensors (Muyldermans et al., 2009).
On the other hand, the chemical constituents of camel urine were first studied by Dr Bernard E. Read in 1925 (Read, 1925). His experimental results described the presence of creatinine, creatine, hippuric acid, total nitrogen ammonia, urea, chlorides and purine bases in urine samples. Noteworthy, only traces of urea and ammonia were found in such samples.
The urine of older camels (5-10 years old) presents a relative density which ranges from 1.01 to 1.07, has variable pH values (acidic or alkaline), urea levels that range from 18 to 36 mg/dl and keratin levels which range from 0.2 to 0.5 mg/l (Amer & Al-Hendi, 1996). Calcium oxalate, phosphorus and ammonium urate were present, as well as some epithelial and granular cells, as revealed through a microscopic study (Amer & Al-Hendi, 1996). In a neutron activation (AI-Attas, 2009), large amounts of Na and K in camel milk and urine analysis carried out by, have which can help reverse the electrolyte imbalance in patients with diarrhoea. High levels of Zn were also identified in the sample. In these regards, zinc has been reported to be an effective remedy against diarrhoeal infections (AI-Attas, 2009).
A recent study explored the unique biochemical constituents of camel urine. The study suggested that camel urine contains slight traces of urea and ammonia, and these molecules are known to be responsible for bad smell and urine toxicity. Additionally, camel urine has been reported to contain approximately 10 times more mineral salts than human urine (Al-Yousef et al., 2012).
Thus, in general terms, the claims of camel urine as a treating element for disease have been tested in several different trials conducted over the years. Still, comparative studies approaching the determination of the properties and constituents of camel’s urine are scarce. This way, further studies evaluating the scientific reasoning that may support camel urine traditional use as a therapeutical agent may help to outlining the idoneity of camel urine (Gole & Hamido, 2020).
In this context, our review gathers and discusses already published documents in high impact journals investigating the therapeutic characteristics of camel urine, the knowledge on its metabolome, and the connection between bioactivity assays and metabolomic studies. The present review has also focused on providing a critical summary of the weaknesses in the methodology used and which may be the main limitans of the reproducibility of the results that have been reported up to present.
For instance, among the weaknesses of studies involving camel urine, the lack of consideration of tolerance limits to the osmolarity of the medium of the cell lines or tissues studied prior to the development of the in vitro/in vivo experiments, which is a key factor when interpreting bioactivity studies and their subsequent transfer to the clinical-pharmaceutical sector. The rationale for this is that camel urine is an organic substrate with generally greater than 1000 mOsm osmolarity levels. Such increased levels cause cell death due to osmotic overload rather than because of the bioactive potential of the molecules it contains.
In addition, other proposals to improve the methodology would be to test for a diverse range of urines from different individuals displaying different phenotypes (sex, age, and other notable morpho-phaneroptic characters) and immerse in diverse environmental contexts (diets, handling, farm policies, among others) which may be potential sources for changes in the chemical constitution of the urine, thus, of its biological activity.
Once the biological activity of whole urine (raw urine) on the cell line(s) of interest has been tested, with urine osmolarity adjusted to the osmotic tolerance of the cell line(s), and considering the animal-dependent variability associated with the variation in the biological activity of each urine, the next step (DIRECTIONS FOR FUTURE RESEARCH) is to analyze the metabolomic profile to know the chemical constitution of the urine and plan in vitro/in vivo studies to test the biological activity of the molecules contained in urine as well as their safety profiles for their use in the formulation of therapeutic drugs (Pastrana et al., 2022).
In short, this is what is called 'Reverse Pharmacology': the science of integrating documented clinical experiences and experiential observations into leads, through transdisciplinary exploratory studies, and further developing these into drug candidates through robust preclinical and clinical research (Patwardhan & Khambholja, 2011). In no case, the objective of this work is to promote the consumption of raw urine, but to standardize an experimental methodology for the sequential and objective evaluation of the potential use of camel urine as a natural source for the discovery and isolation of new molecules. with bioactive potential preserving and promoting Public Health. This is expressed in the text: 'The motivation of the present research is not to promote the consumption of raw camel urine, but to value its role as a source for specific molecules with bioactive potential and that can be safety evaluated in medicinal chemistry and drug discovery in a public health scenario'.
The present methodology is practically the same as that used in Kumar et al. (2021), Redha, Valizadenia, Siddiqui, and Maqsood (2022), Bouglé and Bouhallab (2017), and Morales, Miguel, and Garcés-Rimón (2021), who classified cow urine, camel milk, vegetable and animal derived ingredients present in human diet and pseudocereals contained metabolites, respectively, and sorted them into bioactive groups (anti-microbial, anti-oxidant, anti-inflammatory, anti-diabetic, anti-hypertensive, and anti-cancerous chemicals).
I raise the following questions and remarks:
- In the abstract the authors talk about “studies on the therapeutic properties of camel urine and the detailed metabolomics characterization of its chemical composition are scarce and often unrelated”. What is the scientifically proved medical justification of the use of a waste product of mammalian metabolism to treat humans in the first place?
First, the notable scientific interest in the therapeutic use of camel urine has its origin in the need to verify or refute the safety of an ethnomedical practice that has been patent since ancient civilizations and is still in force today (Mok et al., 2021), with a high symbolic and cultural value, as occurs with an infinity of substances resulting from metabolism or structural constituents in specimens and products of plant and animal origin (Dianita & Jantan, 2017; Kenechukwu et al., 2012; Kulczyński, Sidor, & Gramza-Michałowska, 2019; Okeke, Iroegbu, Eze, Okoli, & Esimone, 2001; Zhang, Chen, & Wang, 2015). In fact, numerous studies have so far reported bioactive effects of camel urine on different pathological processes, although the implemented methodology requires standardization to improve the repeatability and reproducibility of the results, as suggested in this review. However, considering that it is a by-product of organic metabolism, this substrate is likely to generate controversy regarding its consumption in an uncontrolled manner for public health reasons. In this regard, a recent study (Anwar et al., 2021) has reported, either in in vivo and vitro models the absence of genotoxicity after the consumption of raw urine and remarks that camel urine lacks of foul odour due to lower amounts of ammonia and urea, and contains ten-fold mineral salts compared to human urine (Ali, Baby, & Vijayan, 2019). In-vivo mammalian genotoxicity studies are recommended by international drug regulatory authorities for a product to be used in humans to achieve human health safety and for registration as a therapeutic agent (de Souza Marques, Salles, & Maistro, 2015). However, given the possible persistence of discrepancies at this level, the determination of the possibility to efficiently and safely use this by-product for the promotion and improvement of public health requires objective and sequential exploration, according to standardized methodologies in studies previous in the same academic field (Kumar et al. (2021), Redha et al. (2022), Bouglé and Bouhallab (2017), and Morales et al. (2021)), of the substances that have bioactive potential and are contained in camel urine. This way, if these bioactive components are isolated, they could be considered in pharmacological studies whose ultimate goal is the adapted formulation of therapeutic drugs. In this way, the bioactive potential substantially contained in camel urine is contrasted, without the need to expose experimental subjects to possible risks hitherto unknown that would derive from the consumption of raw urine. This is precisely the case of the experimental approach proposed in this review: in no case is the consumption of raw urine being promoted, but rather replicating the methodology used in other studies using different substrates of vegetable and animal origin to identify the metabolites contained in this specific substrate that will be susceptible to isolated research (in-silico and functional tests) to define their bioactivity patterns, safety profiles and potential use in the formulation of new therapeutic drugs. Closely related, we can mention the relatively recent patents dealing with the exploration of certain bioactive fractions obtained from camel urine (AlAttas & Khorshid, 2015; Faten A Khorshid, Osman, & Abdel-Sattar, 2009), as it has also occurred in cow urine (Mahajan, Chavan, Shinde, & Narkhede, 2020), which have been reported to have no toxic effect and found to be safe for human use after phase I studies (F. Khorshid et al., 2015; Osman, 2010). Similarly, human urine has also been used for the treatment of allergic diseases and autoimmune, by the collection of the patient's urine and processing it to obtain the peptide fraction of the urine; this was then administered to the patient after conditioning (Márquez-García et al., 2021).
- The statement “camel urine intra and interindividual variability in terms of chemical composition” is not explained scientifically with the animals’ metabolism, but the authors rely solely on differences between the analytical techniques used. This is a general misunderstanding of physiology.
Response: We understand the reviewer concern. However, the statement which is found in the abstract, is introduced as a proposal for the future, because with the information that is available right now, as discussed on different occasions across the review, the variability of the samples used is too scarce to be able to statistically study the influence of animal-dependent factors on urine metabolomic profiles.
Therefore, by increasing the size and variability of the samples, and if the analytical methodologies are standardized, as proposed in the manuscript, the variability associated with the analytical techniques may be reduced, so that the highest percentage of variability may be associated with dependent factors. of the animal. Previous studies have reported effects of sex (Rettenbacher, Möstl, Hackl, Ghareeb, & Palme, 2004; Touma, Sachser, Möstl, & Palme, 2003), age (Poureslami, Turchini, Raes, Huyghebaert, & De Smet, 2010), diet (Al-Awadi & Al-Judaibi, 2014; Morrow, Kolver, Verkerk, & Matthews, 2002), environmental seasonality (Kuntz, Kubalek, Ruf, Tataruch, & Arnold, 2006; McNab, 2002), composition of gut microbiome (Goymann, 2012; Pusateri, Roth, Ross, & Shultz, 1990), and kinetics of the reactions leading to the production of metabolites or the time course of metabolism (Jacobson & Gerig, 1988), en el metabolismo y sus productos.
This information has been specified to follow the reviewer’s suggestion in the body text.
- in Introduction:
- a) lines 40-45: Why the references used to “justify” the statement are historical interpretations of Mesopotamian texts? In these references, it is clearly stated that “it is believed” and “it is assumed that…” I think these publications are not a source of proven medical knowledge. Refs. 2 and 3 are even not in English and quite hard to obtain.
Response: We understand the reviewer concern. However, these references were chosen as a mean to justify the background behind the interest of the scientific community in studying the bioactivity of this organic substrate. The statements 'it is believed' and 'it is assumed that', as in any study that respects the guidelines of the scientific method, are the basic paradigm from which the experimental design necessary for the contrast of the hypothesis of interest. These paradigms contain a basic set of beliefs or assumptions that guide our inquiries for a particular research (Barnes, 2013; Lincoln, Lynham, & Guba, 2011; Rahi, 2017). Given that this is a common practice since ancient times, and even approved to be used in some medical institutions (hospitals) for the treatment of different pathologies (Abuelgasim et al., 2018; Atteiah et al., 2020), the scientific community has to be ready to perform the pertinent adapted research to accept or refute the fact that camel urine is a pharmacological product, as this current use can be compromising public health if this product is not bioactive but toxic (although it is demonstrated, at least, that this product, when consumed orally, has not genotoxic effects (Anwar et al., 2021). In the case it is methodologically proved that camel urine is bioactive, alternative research on this natural substance such as the isolation of chemicals contained in urine and the testing of biological activity of each compound in an isolated manner, can be a valuable methodology that will aid at demonstrating the bioactive potential contained in urine without the need to consume this substance, and consequently, the discovery of new drugs that can be potentially included in the formulation of therapeutic formulas. This is the methodology used for the discovery of bioactive drugs from natural resources (reverse pharmacology) (Patwardhan & Khambholja, 2011). But we cannot say that as camel urine seems to be an Arabian belief system, and as in any belief, people are not ready to refute it or accept that the topic is esoteric or makes no sense.
- b) Lines 48-57 are another historical interpretation, which is neither medically sound, nor came from a medical publication, but rather philological one. How the authors rely on this?
Response: As aforementioned, in line with the topic, we understand the reviewer concer. However, this is one more justification for the background behind the interest of the scientific community in studying the bioactivity of this organic substrate.We are not relying on this, only citing as a part of the ideological background around the issues of the present review manuscript. As stated by Walker (2013), it is clear that ideology affect science, and it exists ideological interferences in the practice of science itself. So the question would be ‘which is the ‘ideologically correct’ science? (Gordin, Grunden, Walker, & Wang, 2013), and the answer is: the ‘ideologically correct’ science would be determined by dominant ideologies of a state or social aggrupation. To avoid as much as possible the negative impacts of the collision between science and dominant ideology at the time carrying out research opportunities in topics that can result controversial for some social groups or currents of thought, it is enough with the justified and objective application of the scientific method to accept or refute a hypothesis (based on social beliefs or assumptions about a particular issue), or just to do a state-of-the-art review to describe the existing level of scientific development in a field, highlight the weaknesses, and define potential new perspectives/approaches (Lynch, 1994). Understanding the sociology of scientific knowledge and the scientific consensus process can prevent the manipulation of scientific uncertainty (Santelli, 2008).
From a merely medical perspective, Patwardhan and Khambholja (2011) states that ‘the present medicinal system is dominated by the Allopathy or western medicine which is prominently taught and practices in most of the countries worldwide. This system is still evolving and during last few decades focus was based on chemical origin of most of the medicines. Thus majority of drugs in current practice are from synthetic origin. Even so, a large number of these synthetic molecules are based directly or indirectly on natural products or phytoconstituents (Vuorela et al., 2004). We need to understand medicines or systems those were existing in use before emergence of current ‘synthetic era’ and visualize the future of medicine and health care in the ‘technology area’ through the use of emerging technologies for the understanding of the medicine that people have used for thousands of decades and developing new drugs. The linkage between ‘the past’ and ‘the future’ of medicine is much more important and can give us ‘new directions’ for better understanding health, disease and possible solutions. Even WHO has published guideline mentioning different requirements for clinical research on Traditional medicines (Organization, 2000).
- c) Line 61: even cancer is a leading health concern, autourotherapy has neither been studied, nor practiced commonly. How do you explain this isolated cultural phenomenon from the viewpoint of evidence-based medicine?
Response: We apologize, this was a mistake, we wanted to refer to autourotherapy (both with autogenous or heterogeneous urinary extracts). Eldor (1997) describes the results derived from different in-vivo trials in which the administration of urine (both autourotherapy and the administration of urine from other living organisms) has therapeutic or preventive effects on different pathological processes in humans and/or animals, namely: urinary infections , gonarthritis, desensitization, endocrinological problems, migraine, pruritus, asthma, urticaria, eczema, psoriasis, acute and subacute glomerulonephritis, experimental ulcers in dogs, lymphoid depletion in intestinal segments, induction of thrombocytosis in peripheral blood and megakaryocytosis in the spleen, and neutralization of the bone-marrow colony-stimulating factor. Specifically, the effects of urine on oncological pathologies are associated with the immunomodulatory activity of urine (Al-Yousef et al., 2012): ‘Unlike the clonal immunotherapy, the urine of the cancer patients contain the many tumor antigens which constitute the tumor. Oral autourotherapy will provide the intestinal lymphatic system the tumor antigens against which they may produce antibodies due to non-self-recognition. These antibodies may be transpierced through the blood stream and attack the tumor and its cells’ (Vaidya et al., 2003; Wu et al., 2020). Although urotherapy is practiced more or less commonly in different countries for mostly cultural and symbolic reasons (Amira, 2010; Christy, 2019; Thakur, 2004), in other cases if it has been extensively studied to the point of developing medical patents, as is the case of camel urine (Fatin A Khorshid, 2020) and cow urine (Mahajan et al., 2020). The corresponding paragraph has been rewritten to improve its contextualization and reading comprehension.
- d) lines 66-68: There is a chemical misinterpretation of the statement. The potassium and magnesium ions are not directly connected to alkaline pH, but are a result of the renal filtration in alkaline urine. Another misconception here is the statement that albuminuria is a normal condition. If albumin in urine would have pharmacological effect, it should be in levels, high enough to be a diagnostic criterion of albuminuria, thus kidney damage. This is not discussed. How do you justify your views?
Response: We understand the reviewer concern. Obviously, the alkalinity/acidity of urine will depend on the relative concentration of all the various chemical molecules contained in it and on their ionization state. Compared with the other mammals including humans, the alkalinity of camel’s urine may be due to high concentrations of salts such as K, Mg, P and Ca; and little amount of uric acid, sodium, creatinine and carbohydrates (Humaid, 2016; Kamalu, Okpe, & Williams, 2003). In these regards, comparatively higher concentrations of salts such as K, Mg, P and Ca, would be making the pH basic (Kaushal et al., 2017; König, Muser, Dickhuth, Berg, & Deibert, 2009).
Concerning the albumin protein, this fact could be explained by the implication of physical exercise in the temporary presence of protein in the urine of healthy individuals. It is known that strenuous exercise increases protein excretion in healthy patients (Heathcote, Wilson, Quest, & Wilson, 2009; Poortmans, 1964). Camels are in traditional systems that include long-distance journeys to search for food resources or for purely import reasons (Mohamed, El-Maaty, Abd El Hameed, & Ali, 2021), so signs of albuminuria are likely to be found in camels. Indeed, an increase of albumin in camels transported by walk or truck till 18 h after arrival has been found (Emeash, Mostafa, Karmy, Khalil, & Elhussiny, 2016).
The corresponding sentence has been corrected to improve its understanding. “However, camel urine alkalinity (high levels of potassium and, magnesium, and albuminous protein, and low concentrations of uric acid, sodium and creatine), unlike other animal urine, may be the source for the comparatively higher historical transcendence of its use” .
On the other hand, the statement that they need to be in high concentrations to exert an effect is a null statement if done predictively, without references to support it. The dose required for each chemical molecule or combination of molecules to exert a bioactive effect will depend on various biochemical factors of the molecule or molecules, as well as factors dependent on the target cell or cells. You are mixing a pharmacological effect with the possibility of being used as a diagnostic biomarker.
- e) lines 72 to 81 deals with a publication of in vitro cytotoxic effect of urine. This is only one publication, from which it is clearly visible that the cytotoxicity described is both osmotic, from the concentrations used, and due to the toxic metabolites, which the urine contains as a waste product. In vivo results are neither found in literature, nor could in vivo cytotoxicity be extrapolated in an animal. This misinterpretation is a general lack of understanding the difference between cell models and animal models to test activity.
Response: In our opinion, we cannot empirically confirm that the dose-dependent cytotoxic effects reported in the aforementioned work are necessarily due to osmotic reasons, because in no case do references appear of the tolerance levels of the cell lines used to the osmolarity of the culture medium. This is precisely one of the weaknesses of the methodology that is criticized in the review and that stands out as a key element to implement in future projects that seek to find the therapeutic value of camel urine through the isolated and detailed valorization of the molecules contained in this substrate (Pastrana et al., 2022). That is, the evidence of cytotoxicity could also be associated with the fact that the potentially bioactive molecules contained in the urine develop patent biological activity, are found in small concentrations in the urine, and the concentrations from which they develop biological activity significantly patent, are evidently more favourable, the higher the dose of urine or urine extract administered.
In any case, we think that the statement 'it is clearly visible' would not be substantiated because we lack information to make a good interpretation. In relation to this, the study in question is a good example of fact to highlight the main methodological weaknesses in this field of applied research (in fact, it is one of the papers included in the sample reviewed): non-consideration of levels of tolerance of the cells tested to the osmolarity of the medium, no previous adjustment of the osmolarity of the urine or urine extracts used to avoid biased interpretations of the results, and no possibility of association between biological activity and metabolomic composition of the urine or extracts employees.
We find difficulty in knowing which toxic metabolites found in the urine extracts used. Reference is made to the fact that the samples have been "characterized" for their chemical composition using 1H-NMR analysis using DMSO as a solvent system, but the data provided is only that relating to the chemical shifts obtained. But in no case have the metabolites that are generating these chemical shifts been identified, for which extensive experience in handling 1D and 2D spectra is required, which in combination with computational methods and resonance databases, allows the identification of metabolites in complex mixtures, provided that the spectra are acquired under similar conditions and with similar magnetic field strength (Emwas et al., 2019).
Additionally, for a molecule to be effective against a certain therapeutic target, it must be 'toxic'. The question is to control what are the doses and conditions of safe use of this molecule so that it exerts selective toxicity against the target and null or controlled towards the rest of the cells/tissues/organs/systems (Block, 2003).
Lastly, in vivo studies of the effects of camel urine on different pathological processes can be found in the literature. In the case of antimicrobial effect, Jausion, Giard, and Martinaud (1935) describes the results derived from in-vivo experiences in which the administration of urine has therapeutic effects on urinary infections in humans. Other results of in vivo experiments with camel urine are included in some of the papers reviewed in the manuscript and also in Eldor (1997) and Alebie, Yohannes, and Worku (2017). However, and as reflected in the conclusions of the review, the need for a greater number of in vivo experiences in both humans and animals is emphasized in order to at least be able to contrast and verify the effects reported so far in vitro.
- f) lines 82-93: Why there is no information, in this section, about the toxicity of metabolites found in urine. It is known from physiology, that if a compound is “precious” for a mammalian organism, it is reabsorbed in the nephron. How do you justify physiologically that “active compounds” are excreted? Moreover, why there is no information on the cytotoxicity of the metabolites as isolated compounds?
Response: There is no detailed information on toxicity in this section because this set of information is remarkably sparse for molecules found in urine. Therefore, in this section, it is specified that ‘However, information on the bioactivity of specific urine metabolites, their potential interactions and safety profiles is very scarce’. When talking about safety profiles, toxicity studies of each of the molecules are implicitly included in isolation (Kubota, Saito, Ono, & Kodama, 2017). One of the main objectives of the review is precisely to establish what metabolites there are and promote their identification in as many urine samples as possible in order to increase the sample variability and the relationship of this index with different intrinsic and extrinsic factors of the animal; what is known about their biological activity, and lay the foundations for the methodological study of the biological activity, toxicity and safe use profiles of each of them, since they are the necessary steps for the definition of new drugs contained in natural products and that have therapeutic potential.
On the other hand, that a molecule is bioactive does not mean that it cannot be excreted. If, indeed, a precious molecule tries to be reabsorbed in the nephron, the entire concentration of this molecule will not be absorbed in the urine as a consequence of factors that affect the rate of filtration and reabsorption at the glomerular level (Kaskel, Kumar, Lockhart, Evan, & Spitzer, 1987; Khanna & Kurtzman, 2006; Margolick et al., 2014), so a part of its concentration will be excreted in the urine. An example is the concentration of urea in the urine of ruminants. It is known that given diets with little protein content, ruminants reabsorb a significant proportion of urea at the glomerular level to use it as an additional source of amino acid synthesis, but this does not mean that urea cannot be found in their excreted urine (Rogers, 1958). En efecto, providing there is normal renal function, urine, by definition, is the elimination of excess to avoid unwanted effects by bioaccumulation (Bulat, Djukić-Ćosić, Maličević, Bulat, & Matović, 2008; Jackson, Wong, Krier, & Riordan, 2005). On the other hand, urea is biologically active (Yonova & Stoilkova, 2004) and is also used as an additive in animal feed (Briggs, 2014).
Besides, speaking of excretions in general, a biologically active molecule can be perfectly excreted. This is the case of toxic substances excreted by different organisms through different organic pathways as a defense mechanism against environmental threats (Blum, 2012; Cuevas, Martins, Rodrigo, Martins, & Costa, 2018; Habermehl, 2012). That the biologically active substance is active on a different organism from the one that excretes it is explained by different factors such as the biochemistry of the membrane receptors, metabolism and defense mechanisms of the different cell types (Durgo, Belščak-Cvitanović, Stančić, Franekić, & Komes, 2012; Joyce, Fabra, Bozkurt, & Pandit, 2021; Pastrana et al., 2022), and that mark the evolutionary ecology in ecosystems with constantly evolving predator-prey relationships.
- g) lines 98-104: even though some analytical methods are reported in the MS, no critical review was given on them, nor there is any kind of detailed chemical information on neither the capabilities of the methods, nor the findings in urine.
Response: Generally, urine samples were extracted with organic solvents. After derivatization with a silylating reagent, the analytes were identified through GC-MS analysis by comparison with the spectra included in the National Institute of Standard and Technology (NIST) based on a comparison of the unknown’s mass spectrum’s peaks to those of the peaks in the library’s spectra (Match factor > 800). These lines, including the pertinent references, have been introduced and further discussed in the text, along the subsection 3.3.
- Review methodology: this is another issue of the MS – why, even though the authors clearly state that a respected search engine as Science Direct gives only one publication on camel urine, results from Google were used as well? The justification that “governmental and institutional reports that are sometimes not published in indexed journals, but which helps to improve the conceptual comprehensiveness of qualitative systematic reviews.” is not scientifically sound. It rather looks as a conspiracy theory. If a study is good, correctly conducted and the results are sound, there is no problem of publishing it in a respectable scientific journal. A general problem with the Internet is that it is full of unproven information. The MS clearly lack the critical conception of that in mind.
Response: Google Scholar and Google Search are considered to be important sources of grey literature, governmental and institutional reports (Haddaway et al. 2015; Hagstrom et al. 2015). In performing our study, we assumed that not all the guidelines have been published in scientific journals. Therefore, although Google Scholar and Google Search have their limitations and should not be used as the only source for systematic reviews, both seemed to be apt for the purposes of some types of qualitative systematic reviews (Piasecki, Waligora, & Dranseika, 2018).
- Lines 173-177: the journal’s IF is not directly connected to medical soundness of the matter. How do you explain that supposed connection in your text?
Response: Journal Impact Factor should be part of a cohort of considerations including productivity and scientific soundness (Eliades & Athanasiou, 2001). Obviously, there are many factors that condition this bibliometric indicator; among others, how soundness-only peer-review is understood by reviewers (Wakeling et al., 2016) (the inherent nature of peers’ bias).
There are opposing views on the SOPR (prepublication soundness only peer review) approach. Some supporters believe that the SOPR approach has the potential to democratize science, to improve the speed and efficiency of peer review, to facilitate open science, to avoid wasteful rounds of submission-rejection and consequently, to accelerate the speed of scholarly communication. The SOPR method is seen by many supporters as more objective than judgments on perceived novelty, potential significance, subsequent usefulness and anticipated future impact which are criticized as being subjective. These proponents pointed out the shortcomings of the traditional system of quality control, including potential editorial biases, subjectivity, inconsistency, poor reliability, and publication delay, and called into question the fairness of this approach. On the other hand, defenders of the traditional system of peer-review believe that it is the best quality assurance approach available, even with its shortcomings. They argue that the quality control of the OAMJs’ (open access mega journals) content lowered acceptance threshold in scientific community and caused information overload through bulk publishing of difficult-to-publish materials and low-quality outputs that have been rejected by highly selective top-tier conventional titles. Moreover, recruiting a large number of peer reviewers who are completely familiar with the concept of SOPR would be a challenging responsibility of OAMJ publishers and editors and may negatively affect the quality filtering process. Interviewing the publishers and editors of OAMJs, it was found that in many cases OAMJ reviewers conduct the review in the same way as the conventional system so, in reality, taking into account novelty and significance as evaluation criteria in their editorial decisions. Therefore, it is not clear to what extent SOPR-based assessment can remove subjectivity from quality control, nullifying it as one of the four key characteristics of OAMJs. We therefore suggest that SOPR not be a sine qua non factor to qualify an OAMJ as such. Soundness is a multidimensional concept and there is no agreement about what constitutes scientifically sound science. Consequently, OAMJs have different methods of implementing SOPR. Moreover, post publication quality indicators as the second stage of quality control in the SOPR approach are still underdeveloped and it is not clear to what extent these article level metrics would be able to provide evidence-based evaluation about the novelty, significant or relevance of published documents in OAMJs (Erfanmanesh & Teixeira da Silva, 2019).
In our case, the topic of this research study is gaining relatively fast growth within the scientific community mostly in the last decades. Hence, its novelty and the need for standardization of related methodologies may be conditioning its impact and dissemination in high-impact multidisciplinary journals that could favour a broader visibility of this research field among the scientific community (González-Alcaide, Llorente, & Ramos, 2016; Wang, 2018). Por este motivo, la presente revision tiene por objeto highlight the methodological weaknesses that are currently present within this research field and thus the need to standardize the methodology for further, reliable development of the research field.
- Table 1: Why there is no statement how many papers, dealing with camel urine, from a medical viewpoint, the authors have found in the first place? For a 96 years’ period if this urine is really effective, it should have been at least several. Why there was a necessity to elongate the search terminology, if the camel urine is so widely used?
Response: In the first search, we obtained a total of 762 references between the 2 databases used (ScienceDirect (141 references) and Google Schoolar (621 references)).
In ScienceDirect, 140 references were related to urine bioactivity and 1 reference to urine metabolome. Now, one of the references related to urine bioactivity was a review that in turn included 6 articles, which we decided to break down to individually review each of the papers included as part of our study sample, making our sample then 146 references. . However, 3 of these 6 papers included in this review were already within the sample to be reviewed selected in the first place (that is, after the first filtering), so their inclusion would generate duplicates. If we remove these duplicates, we are then left with a final sample of 143 references from this database = 142 references related to urine bioactivity and 1 reference related to urine metabolome. Now, discarding from ScienceDirect the references that only cited the topics of interest for the review but were not research studies per se (n=132), we were left with 11 references (10 on urine bioactivity and 1 on urine metabolome).
Applying the same logic in Google Scholar, 614 references were related to urine bioactivity and 7 references to urine metabolome. In Google Schoolar, 614 references were discarded, leaving us with only 7, all related to the urine metabolome.
In short, we are left with 18 references (11 from Science Direct (10 on urine bioactivity and 1 on urine metabolome) and 7 from Google Schoolar (all related to urine metabolome)).
The corresponding paragraph has been corrected in the text to briefly include this information. Also in Figure 1, the statement 'Those documents not versing explicitly on the hot topics/just simply citing them and duplicates (n=746), were discarded' has been corrected.
For a period of 96 years, there are many references but they only cite the topics, they are not research studies as such and therefore we cannot include them in this review. Again, the background of the subject is broad because it is based on a symbolism and cultural heritage of many years ago, but the development of science is being evident, especially in recent decades. Besides, the fact that there are not a considerable number of papers does not mean that this is not a widespread practice and, therefore, worthy of being approached from a scientific point of view to contrast/refute all the surrounding hypotheses regarding the effectiveness of its use and its safety profile in a context of safeguarding and promoting public health. Again, the negative impacts of the prevailing ideologies on the development opportunities of certain scientific fields come into play here.
The need to extend the search terminology is to make the search more precise.
- How do you explain “not indicated” in Table 1? This concerns the species of the camels, the breeding location of the animals, the sample size, their sex and their age. These are variables that in general are responsible for statistical differences between animals and their urine. If you accept articles, in which those are not stated, what is the statistical significance of your review in the first place?
Response: The significance of the review is in the conclusions, this is not a meta-analysis of the therapeutic effects of camel urine. Statistical analysis is purely bibliometric.
- lines 224 to 237 are a general talk on camels’ importance as a livestock and have nothing in connection with neither the topic of the MS, nor with the scope of the Journal.
Response: The fact of highlighting the importance of the camel as a livestock species, its worldwide distribution and its 5 levels of economic importance serves to create in the reader a general idea of the socio-demographic importance of this ethnopharmacological practice but also to get an idea of the potential variability that can be found in the metabolic composition of camel urine due to the enormous variability in geography, genetics and husbandry practices between the different areas where these animals are present.
- lines 263-260: How do you explain your views, taking in mind my remark of the same (see 3e, above)?
Response: I do not understand this question very well, but if I have understood it correctly, the answer is the same as that detailed in subsection 3e of the review comments. The misinterpretation has been made by the reviewer, because he has drawn conclusions not based on the information present in the cited paper. It is a paper published in the Journal of Ethnopharmacology, dedicated to the exchange of information and understandings about people's use of plants, fungi, animals, microorganisms and minerals and their biological and pharmacological effects based on the principles established through international conventions. The fact that it is an article with a considerable number of citations does not mean that it is good, obviously. The weaknesses in methodology are already explained in subsection 3e of these comments, and they are one of the main weaknesses criticized at a general level in this review.
- Paragraph 3.2: This is a statistical interpretation of publications found. It is neither an evidence-based, nor statistically sound connection between the effect stated and the sample size. How do you justify statistically results, accumulated in less than five papers, some of which even written in non-medical and non-chemically oriented journals?
Response: It is a summary of the current status of knowledge on the bioactive effects of the camel urine aimed to remark the methodological weaknesses and propose how to perform future research in this field so that reproducibility and replicability are scientifically evidenced improved.
- There is no critical interpretation of lines 498 to 502. Neither the oral dose of urine is justified in the reference stated, nor is the water immersion-restraint stress (WRS) model an approved and fully working model.
Response: We have included a critical interpretation of those paragraphs in the text in which experimental doses are suggested: ‘However, the up-to-present suggested as effective doses of camel urine, cannot be trustful since the methodological approaches have been a single run instead of an iterative process (five steps or phases), which is the accepted standardized strategy for drug development in medicinal chemistry (Ariëns, 2017)’.
Concerning the methodology of Salamt et al. (2021), water immersion restraint stress (WRS) mimics the clinical acute gastric ulcerations caused by trauma, surgery, or sepsis and has been widely accepted for studying stress ulceration. It is theoretically and clinically significant to demonstrate the mechanism of stress-induced gastric injury and develop respective therapeutic drugs (Guo et al., 2012).
- Why there is no information on the biochemical possibility of some organic compounds to be found in the urine, or their eventual toxicity in Table 3?
Response: Relevant biological information was reported in Table S2. A critical assessment of the relevance of the compounds found in urine as drugs or fragments was initially reported in lines 386-404: ‘As expected, many of the enlisted metabolites are relatively hydrophilic with their corresponding calculated partition coefficient (cLogP) values often being ≤ 0 (Figure 3, panel A; Table S2). This should prevent any systemic pharmacological effect for most of them since cLogP values of pharmacologically relevant compounds generally fall within the 0 to 3 cLogP range. The inference is in agreement with the very low drug-likeness shown by most of the major constituents in camel urine (cf. Table S2 and panels B and C in Figure 3). A fraction ≥ 75% of the stated metabolites should be endowed with good oral bioavailability since they do not violate the ‘Rule of Five’ (Ro5) test and Veber’s filter for oral absorption (Table S2 and panels D and E in Figure 3). The major components found in camel urine are small molecules (MW = 200±100; Table S2) and 38% of them could be considered as useful fragments to develop rather complex derivatives with good expectations for biological activity since they fulfil the ‘Rule of Three’ (Ro3) proposed for fragment-based lead discovery (panel F in Figure 3). The quality of those fragments is high in most cases as it may be inferred from the high fraction of sp3 hybridized carbon atoms (Fsp3) found in most of them and the frequent presence of chirality centres. High Fsp3 and chirality are positively related to the rate of success in clinical development (Panels G and H in Figure 3, and Table S2)’.
Even though compound as thiophene, pyridine, quinolone and even phenol are presented as “found’ in urine, there is no critical review neither of this possibility, nor of the possible health damage that they would induce in an organism after ingestion.
Thiophene, pyridine, furan,1H-pyrrole, pyridine, and quinoline were erroneously included in Table 3 and were left out in the amended version. Careful inspection of the analytical methods (generally roughly described in most of the cited papers) compelled the exclusion of several compounds that were “found” in urine while being probably food contaminants or artefactual derivatives (Si-containing compounds). Only organic compounds were reported, and the heavy metal complexes reported in the original papers as “found” were excluded. Consequently, the graphs in Figure 3 were corrected since they derive from a reduced number of compounds now (72 in the amended version; 101 in the previous one).
Moreover, the statement that Acetic acid was found in urine, even though the authors clearly state that camel urine is alkaline is a very strange understanding of chemistry.
We agree with the reviewer: acetic acid, like all carboxylic acids, is mostly in its corresponding dissociated form in alkaline solutions. Thus, we should refer to the acetate (and more generally to carboxylate) anion(s) in those instances. However, when a list of compounds is to be reported, generally we refer to the non-ionized forms of both acidic and basic compounds.
Reviewer 2 Report
The authors have written a systematic review of the biological properties of camel urine. The discussion on pharmacological studies should include information on the amount dosed to achieve an effect. Section 3.3 must be carefully checked, as MS will detect impurities and contaminants from sample processing. The table, for example, contains compounds that are certainly not present in camel urine such as triethyl(fluoro)silane and cyclooctanone. The authors should consult with experts in MS in interpreting the data.
Author Response
Reviewer 2
The authors have written a systematic review of the biological properties of camel urine. The discussion on pharmacological studies should include information on the amount dosed to achieve an effect. Section 3.3 must be carefully checked, as MS will detect impurities and contaminants from sample processing. The table, for example, contains compounds that are certainly not present in camel urine such as triethyl(fluoro)silane and cyclooctanone. The authors should consult with experts in MS in interpreting the data.
We agree with the reviewer. Careful inspection of the analytical methods (generally roughly described in most of the cited papers) compelled the exclusion of several compounds that were “found” in urine while being probably food contaminants, artifactual derivatives, or compounds coming from partial degradation of the stationary phases (Si-containing compounds). Only organic compounds were reported, and the heavy metal complexes reported in the original papers as “found” were excluded. Consequently, the graphs in Figure 3 were corrected since they derive from a reduced number of compounds now (72 in the amended version; 101 in the previous one).
On the other hand, we cannot include information on the effective dose because we are not promoting the use of raw urine as a therapeutic alternative, but rather proposing a different approach for the isolated study of the molecules contained in camel urine that could potentially be bioactive and, therefore, used in the formulation of therapeutic drugs. In other words, the objective of this review is to propose the standardization of the methodology in this field of applied research as well as a future proposal to empirically contrast the bioactive potential contained in this organic substrate for the isolation of new potentially therapeutic drugs.
Once the chemicals contained are known, isolated pharmacological studies will be carried out for each of them, to determine, among other things, the effective and safe dose of each of them.
This is precisely the proposal for standardization and future approach of the methodology that is proposed in the review. With regard to the information on the amount dosed to achieve an effect, only the literary references that specifically speak of 'effective doses' are cited. However, this is a knowledge gap within the topic. As indicated in the revised version of the manuscript just after the citation of these references: 'However, the up-to-present suggested as effective doses of camel urine, cannot be trustful since the methodological approaches have been a single run instead of an iterative process (five steps or phases), which is the accepted standardized strategy for drug development in medicinal chemistry (Ariëns, 2017)’.
Round 2
Reviewer 1 Report
The authors failed to give key medical, physiological and chemical answers to the raised questions before. Although some very long answers were written, the MS is neither improved from a medical, nor from a chemical viewpoint. There is a lack of medical understanding on the matter of "therapy with a waste fluid".